# The Interaction of Selective A1 and A2A Adenosine Receptor Antagonists with Magnesium and Zinc Ions in Mice: Behavioural, Biochemical and Molecular Studies

**DOI:** 10.3390/ijms22041840

**Published:** 2021-02-12

**Authors:** Aleksandra Szopa, Karolina Bogatko, Mariola Herbet, Anna Serefko, Marta Ostrowska, Sylwia Wośko, Katarzyna Świąder, Bernadeta Szewczyk, Aleksandra Wlaź, Piotr Skałecki, Andrzej Wróbel, Sławomir Mandziuk, Aleksandra Pochodyła, Anna Kudela, Jarosław Dudka, Maria Radziwoń-Zaleska, Piotr Wlaź, Ewa Poleszak

**Affiliations:** 1Chair and Department of Applied and Social Pharmacy, Laboratory of Preclinical Testing, Medical University of Lublin, 1 Chodźki Street, PL 20–093 Lublin, Poland; karolina.bogatko@umlub.pl (K.B.); anna.serefko@umlub.pl (A.S.); sylwia.wosko@umlub.pl (S.W.); 2Chair and Department of Toxicology, Medical University of Lublin, 8 Chodźki Street, PL 20–093 Lublin, Poland; mariola.herbet@umlub.pl (M.H.); marta.ostrowska@umlub.pl (M.O.); ankaku@poczta.onet.pl (A.K.) jaroslaw.dudka@umlub.pl (J.D.); 3Chair and Department of Applied and Social Pharmacy, Medical University of Lublin, 1 Chodźki Street, PL 20–093 Lublin, Poland; katarzyna.swiader@umlub.pl (K.Ś.); olaap@onet.pl (A.P.); 4Department of Neurobiology, Polish Academy of Sciences, Maj Institute of Pharmacology, 12 Smętna Street, PL 31–343 Kraków, Poland; szewczyk@if-pan.krakow.pl; 5Department of Pathophysiology, Medical University of Lublin, 8 Jaczewskiego Street, PL 20–090 Lublin, Poland; aleksandra.wlaz@umlub.pl; 6Department of Commodity Science and Processing of Raw Animal Materials, University of Life Sciences, 13 Akademicka Street, PL 20–950 Lublin, Poland; piotr.skalecki@up.lublin.pl; 7Second Department of Gynecology, 8 Jaczewskiego Street, PL 20–090 Lublin, Poland; wrobelandrzej@yahoo.com; 8Department of Pneumology, Oncology and Allergology, Medical University of Lublin, 8 Jaczewskiego Street, PL 20–090 Lublin, Poland; slawomir.mandziuk@umlub.pl; 9Department of Psychiatry, Medical University of Warsaw, 27 Nowowiejska Street, PL 00–665 Warsaw, Poland; maria.radziwon@wum.edu.pl; 10Department of Animal Physiology and Pharmacology, Institute of Biological Sciences, Maria Curie–Skłodowska University, Akademicka 19, PL 20–033 Lublin, Poland; piotr.wlaz@umcs.lublin.pl

**Keywords:** DPCPX, istradefylline, magnesium, zinc, antidepressant activity

## Abstract

The purpose of the study was to investigate whether the co-administration of Mg^2+^ and Zn^2+^ with selective A1 and A2A receptor antagonists might be an interesting antidepressant strategy. Forced swim, tail suspension, and spontaneous locomotor motility tests in mice were performed. Further, biochemical and molecular studies were conducted. The obtained results indicate the interaction of DPCPX and istradefylline with Mg^2+^ and Zn^2+^ manifested in an antidepressant-like effect. The reduction of the BDNF serum level after co-administration of DPCPX and istradefylline with Mg^2+^ and Zn^2+^ was noted. Additionally, Mg^2+^ or Zn^2+^, both alone and in combination with DPCPX or istradefylline, causes changes in *Adora1* expression, DPCPX or istradefylline co-administered with Zn^2+^ increases *Slc6a15* expression as compared to a single-drug treatment, co-administration of tested agents does not have a more favourable effect on *Comt* expression. Moreover, the changes obtained in *Ogg1*, *MsrA*, *Nrf2* expression show that DPCPX-Mg^2+^, DPCPX-Zn^2+^, istradefylline-Mg^2+^ and istradefylline-Zn^2+^ co-treatment may have greater antioxidant capacity benefits than administration of DPCPX and istradefylline alone. It seems plausible that a combination of selective A1 as well as an A2A receptor antagonist and magnesium or zinc may be a new antidepressant therapeutic strategy.

## 1. Introduction

The magnesium ion (Mg^2+^), a fundamental intracellular cation, and the zinc ion (Zn^2+^), the second most abundant trace element in the human body, are indispensable for the proper course of various physiological processes [1,2], including the central nervous system (CNS) functioning and development [1,3]. Both preclinical and clinical research conducted over recent years have provided a great deal of evidence for the involvement of Mg^2+^ and Zn^2+^ in the etiopathogenesis and therapy of depressive disorders. It is mainly related to disturbances of glutamatergic transmission in brain structures that play a key role in the pathophysiology of depression [4,5,6,7,8]. Additionally, magnesium and zinc enhance the activity of antidepressant agents (e.g., imipramine, fluoxetine, paroxetine, citalopram, tianeptine and bupropion) [4,5,9,10,11,12].

Currently, the best documented mechanism of antidepressant-like activity of magnesium ions seems to be their inhibitory effect on glutamatergic transmission through antagonism of *N*–methyl–ᴅ–aspartic (NMDA) receptor complex [13]. It was shown that the antidepressant activity of magnesium recorded in the forced swim test (FST) was antagonized by NMDA receptor agonist (NMDA or ᴅ–serine) co-administration [14]. Moreover, Poleszak et al. [14] demonstrated that Mg^2+^ potentiates the antidepressant-like activity of NMDA receptor complex antagonists (L–701,324, CGP 37849, dizocilpine, ᴅ–cycloserine) in a mouse despair test when used jointly at non-active doses. The leading mechanism of Zn^2+^ action, similarly to Mg^2+^, is its effect on the activity of the NMDA receptor complex. Synaptic zinc modulates also activity of other ionotropic glutamate receptors, i.e., α–amino–3–hydroxy–5–methyl–4–isoxazole–propionic acid (AMPA) receptors, as well as metabotropic glutamate receptors, mGluR. Moreover, Zn^2+^ affects functions of γ–aminobutyric acid (GABA–A), glycine inotropic as well as GPR39 metabotropic receptors (a specific Zn^2+^–susceptible receptor) [13]. In the literature two distinct zinc mechanisms of NMDA receptor complex inhibition have been described: a non-competitive and voltage–independent antagonism, as a result of which the opening frequency of the receptor channel is decreased, and voltage–dependent antagonism contributing to blocking the receptor channel opening [15,16,17].

Adenosine is a significant endogenous neuromodulator in the CNS [18]. Adenosine receptors play pivotal role in mediating adenosine transduction [19,20]. Regarding biochemical, pharmacological, structural functions and properties, four subtypes of adenosine receptors have been distinguished, i.e., A1, A2A, A2B and A3 [19]. Among them, A1 and A2A receptors are the most abundant in the brain. The A1 receptor is coupled to G_i/o_ protein and is distributed all over the CNS (i.e., cerebral cortex, hippocampus, striatum, thalamic, cerebellum, the dorsal part of the spinal cord). Its stimulation entails inhibition of adenylate cyclase and Ca^2+^ channels, but activation of K^+^ channels [21]. Instead, the A2A receptor is coupled to G_s_ protein and enhances the adenylate cyclase activity. This subtype of adenosine receptors is mostly localized in the dorsal striato-pallidal GABA pathway and also in the nucleus accumbens, cerebral cortex and hippocampus [18,19]. Furthermore, in the striatal glutamate nerve terminals at the presynaptic level A1–A2A heterotertameric receptor complexes were recognized [20,22]. 

A growing body of evidence indicate the interaction between the glutamatergic and adenosine system, and there are several possible mechanisms of association between these systems. Firstly, by stimulating the A1 and A2A receptors, adenosine modulates the release of several neurotransmitters, including glutamate [23]. Additionally, adenosine by antagonizing membrane depolarization elevates the threshold needed to open the NMDA receptor channels [24]. Moreover, it has been shown that relationship between NMDA and A1 receptors leads to down-regulation of presynaptic glutamate release in neurons of the cingulate cortex [25], hippocampus [26] and striatum [27]. It was also presented that elevate extracellular concentration of adenosine could induce the A2A receptor protomer in the A1–A2A heteroreceptor complex exhibiting an antagonistic allosteric receptor-receptor interaction restricting A1 receptor protomer signalling. This receptor complex is localized on the striatal glutamate nerve terminals and the reduction of the inhibitory A1 receptor protomer signalling causes enhancement in glutamate release [28]. However, to the best of authors’ knowledge, the interaction between selective adenosine A1 or A2A receptor antagonists and ionic NMDA receptor antagonists in behavioural tests in rodents has not been studied hitherto. Since the antidepressant effect of Mg^2+^ and Zn^2+^ ions [9,10] and both non-selective [29,30] and selective [31,32,33] antagonists of the adenosine receptors in mice despair tests have been proven, we decided to examine whether the co-administration of magnesium and zinc hydroaspartate with the selective A1 or A2A receptor antagonist (DPCPX and istradefylline, respectively) might be an interesting strategy in the context of depression therapy. To this end, we have performed behavioural tests (FST, tail suspension test (TST) and spontaneous locomotor motility test) as well as biochemical and molecular studies in which we evaluated: (1) The serum level of brain–derived neurotrophic factor (BDNF), which is common neurotrophic factors in adult humans and animals and is acknowledged as one of the biomarkers of depression, and (2) the expression of selected genes that may play a role in the pathophysiology and treatment of depression or the mechanism of adenosine A1 and A2A antagonists action (i.e., *Adora1, Slc6a15, Comt*), and (3) the expression of selected antioxidant defence genes (i.e., *Ogg1*, *MsrA*, *Nrf2*) since oxidative stress plays an important role in the aetiology of depression.

## 2. Results

### 2.1. Behavioural Studies

#### 2.1.1. Effect of Co-Administration of Selective Adenosine Receptor Antagonists and Magnesium or Zinc in the FST

(1)DPCPX and Magnesium or Zinc

As shown in Figure 1A neither DPCPX (1 mg/kg) nor Mg^2+^ (10 mg/kg) nor Zn^2+^ (2.5 mg/kg) caused statistically significant changes in the FST (*p* > 0.05).

DPCPX and Mg^2+^ injected simultaneously at non-effective doses (1 and 10 mg/kg, respectively) caused a significant decrease in total immobility time in comparison to NaCl-, DPCPX- and Mg^2+^-treated group (*p* < 0.05, *p* < 0.01 and *p* < 0.01, respectively). A two-way ANOVA showed a significant interaction between DPCPX and Mg^2+^ [F(1,35) = 6.61, *p* = 0.0145].

DPCPX and Zn^2+^ injected simultaneously at non-effective doses (1 and 2.5 mg/kg, respectively) caused a significant decrease in total immobility time in comparison to DPCPX- and Zn^2+^-treated group (*p* < 0.05). A two-way ANOVA showed a significant interaction between DPCPX and Zn^2+^ [F(1,35) = 6.45, *p* = 0.0157].

(2)Istradefylline and Magnesium or Zinc

As shown in Figure 1A neither istradefylline (0.5 mg/kg) nor Mg^2+^ (10 mg/kg) nor Zn^2+^ (2.5 mg/kg) caused statistically significant changes in the FST (*p* > 0.05).

Istradedylline and Mg^2+^ injected simultaneously at non-effective doses (0.5 and 10 mg/kg, respectively) caused a significant decrease in total immobility time in comparison to NaCl-, istradefylline- and Mg^2+^-treated group (*p* < 0.0001). A two-way ANOVA showed a significant interaction between istradefylline and Mg^2+^ [F(1,36) = 20.76, *p* < 0.0001].

Istradefylline and Zn^2+^ injected simultaneously at non-effective doses (0.5 and 2.5 mg/kg, respectively) caused a significant decrease in total immobility time in comparison to NaCl-, istradefylline- and Zn^2+^-treated group (*p* < 0.0001). A two-way ANOVA showed a significant interaction between istradefylline and Zn^2+^ [F(1,36) = 18.78, *p* = 0.0001].

#### 2.1.2. Effect of Co-Administration of Selective Adenosine Receptor Antagonists and Magnesium in the TST

(1)DPCPX and Magnesium or Zinc

As shown in Figure 1B neither DPCPX (1 mg/kg) nor Mg^2+^ (10 mg/kg) nor Zn^2+^ (2.5 mg/kg) caused statistically significant changes in the TST (*p* > 0.05).

DPCPX and Mg^2+^ injected simultaneously at non-effective doses (1 and 10 mg/kg, respectively) caused a significant decrease in total immobility time in comparison to NaCl-, DPCPX- and Mg^2+^-treated group (*p* < 0.0001). A two-way ANOVA showed a significant interaction between DPCPX and Mg^2+^ [F(1,36) = 14.73, *p* = 0.0005].

DPCPX and Zn^2+^ injected simultaneously at non-effective doses (1 and 2.5 mg/kg, respectively) caused a significant decrease in total immobility time in comparison to NaCl-, DPCPX- and Zn^2+^-treated group (*p* < 0.0001). A two-way ANOVA showed a significant interaction between DPCPX and Zn^2+^ [F(1,36) = 13.76, *p* = 0.0007].

(2)Istradefylline and Magnesium or Zinc

As shown in Figure 1B neither istradefillyne (0.5 mg/kg) nor Mg^2+^ (10 mg/kg) nor Zn^2+^ (2.5 mg/kg) caused statistically significant changes in the FST (*p* > 0.05).

Istradefylline and Mg^2+^ injected simultaneously at non-effective doses (0.5 and 10 mg/kg, respectively) caused a significant decrease in total immobility time in comparison to NaCl-, istradefylline- and Mg^2+^-treated group (*p* < 0.0001). A two-way ANOVA showed a significant interaction between istradefylline and Mg^2+^ [F(1,36) = 23.98, *p* < 0.0001].

Istradefylline and Zn^2+^ injected simultaneously at non-effective doses (0.5 and 2.5 mg/kg, respectively) caused a significant decrease in total immobility time in comparison to NaCl-, istradefylline- and Zn^2+^-treated group (*p* < 0.0001). A two-way ANOVA showed a significant interaction between istradefylline and Zn^2+^ [F(1,36) = 22.46, *p* < 0.0001].

#### 2.1.3. Spontaneous Locomotor Motility

The effect of DPCPX (1 mg/kg), istradefylline (0.5 mg/kg), Mg^2+^ (10 mg/kg), Zn^2+^ (2.5 mg/kg) and co-administration of Mg^2+^ or Zn^2+^ with DPCPX or istradefylline on spontaneous locomotor motility in mice is shown in Table 1. DPCPX, istradefylline, and Mg^2+^ given alone or in combination had no statistically significant effects on locomotor motility in mice (*p >* 0.05). Zn^2+^ administered alone and in combination with DPCPX significantly decreases the distance travelled by mice (*p <* 0.05 and *p <* 0.001 in comparison to NaCl-treated group, respectively).

The two-way ANOVA demonstrated: (1) no interaction between DPCPX and Mg^2+^ [F(1,33) = 0.08, *p* = 0.7833], (2) no interaction between DPCPX and Zn^2+^ [F(1,34) = 0.17, *p* = 0.6847], (3) no interaction between istradefylline and Mg^2+^ [F(1,33) = 3.94, *p* = 0.0555], (4) a significant interaction between istradefylline and Zn^2+^ [F(1,34) = 5.25, *p* = 0.0283].

### 2.2. Biochemical and Molecular Studies

#### 2.2.1. Effect of Co-Administration of Selective Adenosine Receptor Antagonists and Magnesium or Zinc on BDNF Concentration

(1)DPCPX and Magnesium or Zinc

As shown in Figure 2 neither DPCPX nor Mg^2+^ nor Zn^2+^ injected alone caused statistically significant changes in BDNF concentration (*p* > 0.05) as compared to the control group. Additionally, in the groups of animals that received DPCPX in a combination with Zn^2+^ no statistically significant changes in the concentration of BDNF were observed (*p* > 0.05). However, the co-administration of DPCPX with Mg^2+^ resulted in a significant decrease in BDNF level as compared to the control group (*p* < 0.0001) and to the group receiving DPCPX alone (*p* < 0.0001).

A two-way ANOVA showed no interaction between DPCPX and Mg^2+^ [F(1,20) = 4.20, *p* = 0.0538] and no interaction between DPCPX and Zn^2+^ [F(1,20) = 0.01, *p* = 0.9180].

(2)Istradefylline and Magnesium or Zinc

As shown in Figure 2 neither istradefylline nor Mg^2+^ nor Zn^2+^ caused statistically significant changes in BDNF concentration (*p* > 0.05) as compared to the control group. The co-administration of istradefylline with Mg^2+^ resulted in a significant decrease in BDNF level as compared to the control group (*p* < 0.0001) and to the group receiving istradefylline or Mg^2+^ alone (*p* < 0.05). Additionally, in the groups of animals that received istradefylline in combination with Zn^2+^, a statistically significant decrease in the BDNF concentration in comparison to the control group and to the group receiving istradefylline alone was observed (*p* < 0.01 and *p* < 0.05, respectively).

A two-way ANOVA showed no interaction between istradefylline and Mg^2+^ [F(1,20) = 0.75, *p* = 0.3959] and no interaction between istradefylline and Zn^2+^ [F(1,20) = 0.14, *p* = 0.7086].

#### 2.2.2. Effect of Co-Administration of Selective Adenosine Receptor Antagonists and Magnesium or Zinc on *Adora1* Expression

(1)DPCPX and Magnesium or Zinc

As shown in Figure 3A DPCPX administered alone did not cause significant changes in the expression of *Adora1*. However, the single administration of Mg^2+^ or Zn^2+^ resulted in a significant increase in *Adora1* mRNA level as compared to the control group (*p* < 0.01 and *p* < 0.001, respectively). Interestingly, DPCPX co-administered with Mg^2+^ caused a significant decrease in the *Adora1* expression in comparison to the control group (*p* < 0.0001) and also to the DPCPX-treated group and to the group receiving Mg^2+^ alone (*p* < 0.0001). In turn, simultaneous treatment with DPCPX with Zn^2+^ resulted in a significant increase in *Adora1* mRNA level as compared to the group that received DPCPX or Zn^2+^ (*p* < 0.0001) alone.

A two-way ANOVA showed a significant interaction between DPCPX and Mg^2+^ [F(1,56) = 36.29, *p* < 0.0001] and a significant interaction between DPCPX and Zn^2+^ [F(1,56) = 122.56, *p* < 0.0001].

(2)Istradefylline and Magnesium or Zinc

As shown in Figure 3A istradefylline injected mice alone did not cause significant changes in the expression of Adora1. The single administration of Mg^2+^ or Zn^2+^ and the co-administration of istradefylline with Mg^2+^ to mice resulted in a significant increase in Adora1 mRNA level as compared to the control group.

In the groups of animals that received istradefylline in a combination with Zn^2+^ a statistically significant increase in Adora1 mRNA level was observed as compared to the control group (*p* < 0.0001) and to the group received Zn^2+^ alone (*p* < 0.001).

A two-way ANOVA showed no interaction between istradefylline and Mg^2+^ [F(1,80) = 3.06, *p* = 0.0839] and no interaction between istradefylline and Zn^2+^ [F(1,80) = 0.42, *p* = 0.5202].

#### 2.2.3. Effect of Co-Administration of Selective Adenosine Receptor Antagonists and Magnesium or Zinc on Slc6a15 Expression

(1)DPCPX and Magnesium or Zinc

As shown in Figure 3B Mg^2+^ and Zn^2+^ injected alone caused a significant decrease the mRNA level of *Slc6a15* in comparison to the control group (*p* < 0.05 and *p* < 0.0001, respectively), while DPCPX did not cause statistically significant changes in *Slc6a15* gene expression (*p* > 0.05). Additionally, DPCPX co-administered with Mg^2+^ did not cause significant changes in the expression of this gene. However, the co-administration of DPCPX with Zn^2+^ resulted in a significant increase in *Slc6a15* mRNA levels as compared to the group receiving only Zn^2+^ (*p* < 0.0001).

A two-way ANOVA showed no interaction between DPCPX and Mg^2+^ [F(1,56) = 0.56, *p* = 0.4584] and a significant interaction between DPCPX and Zn^2+^ [F(1,56) = 17.77, *p* < 0.0001].

(2)Istradefylline and Magnesium or Zinc

As shown in Figure 3B both istradefylline and Mg^2+^ as well as Zn^2+^ injected alone caused a significant decrease in the mRNA level of *Slc6a15* in comparison to the control group (*p* < 0.0001, *p* < 0.05 and *p* < 0.0001, respectively). Istradefylline and Mg^2+^ administered simultaneously caused a significant decrease in the *Slc6a15* expression in comparison to the control group (*p* < 0.0001) and a significant increase in comparison to the istradefylline-treated group (*p* < 0.0001). However, istradefylline co-administered with Zn^2+^ resulted in a significant increase in *Slc6a15* mRNA levels as compared to the istradefylline-treated group (*p* < 0.0001) as well as Zn^2+^-treated group (*p* < 0.0001).

A two-way ANOVA showed a significant interaction between istradefylline and Mg^2+^ [F(1,80) = 24.26, *p* < 0.0001] and a significant interaction between istradefylline and Zn^2+^ [F(1,80) = 81.06, *p* < 0.0001].

#### 2.2.4. Effect of Co-Administration of Selective Adenosine Receptor Antagonists and Magnesium or Zinc on Comt Expression

(1)DPCPX and Magnesium or Zinc

As shown in Figure 3C both DPCPX and Mg^2+^ as well as Zn^2+^ administered alone caused a significant increase in the mRNA level of *Comt* in comparison to the control group (*p* < 0.05, *p* < 0.01 and *p* < 0.0001, respectively). However, DPCPX co-administered with Mg^2+^ caused a significant decrease in the *Comt* expression in comparison to the control group (*p* < 0.01), to the to the DPCPX-treated group (*p* < 0.0001) and also to the Mg^2+^-treated group (*p* < 0.0001). Similarly, the simultaneously administration to mice the DPCPX with Zn^2+^ resulted in a significant reduction in *Comt* mRNA levels both compared to the DPCPX-treated group (*p* < 0.001) and to the group receiving only Zn^2+^ (*p* < 0.0001).

A two-way ANOVA showed a significant interaction between DPCPX and Mg^2+^ [F(1,56) = 35.53, *p* < 0.0001] and a significant interaction between DPCPX and Zn^2+^ [F(1,56) = 53.64, *p* < 0.0001].

(2)Istradefylline and Magnesium or Zinc

As shown in Figure 3C both istradefylline and Mg^2+^ as well as Zn^2+^ injected alone caused a significant increase in the mRNA level of *Comt* in comparison to the control group (*p* < 0.01, *p* < 0.01 and *p* < 0.0001, respectively). Istradefylline and Mg^2+^ administered simultaneously caused a significant decrease in the *Comt* expression in comparison to the groups received alone istradefylline or Mg^2+^ (*p* < 0.01 and *p* < 0.0001, respectively). In turn, the co-administration of istradefylline with Zn^2+^ to mice resulted in a significant increase in *Comt* mRNA levels as compared to the control group (*p* < 0.01) and a significant decrease as compared to the group received only istradefylline or Zn^2+^ (*p* < 0.05 and *p* < 0.001, respectively).

A two-way ANOVA showed a significant interaction between istradefylline and Mg^2+^ [F(1,80) = 27.27, *p* < 0.0001] and a significant interaction between istradefylline and Zn^2+^ [F(1,80) = 23.71, *p* < 0.0001].

#### 2.2.5. Effect of Co-Administration of Selective Adenosine Receptor Antagonists and Magnesium or Zinc on Ogg1 Expression

(1)DPCPX and Magnesium or Zinc

As shown in Figure 3D neither DPCPX nor Mg^2+^ nor Zn^2+^ caused statistically significant changes in *Ogg1* gene expression (*p* > 0.05) as compared to the control group. Additionally, DPCPX co-administered with Mg^2+^ did not cause significant changes in the expression of this gene. However, the co-administration of DPCPX with Zn^2+^ resulted in a significant reduction in *Ogg1* mRNA levels both compared to the control group (*p* < 0.0001) and to the group receiving only DPCPX (*p* < 0.0001).

A two-way ANOVA showed no interaction between DPCPX and Mg^2+^ [F(1,56) = 0.96, *p* = 0.3323] and a significant interaction between DPCPX and Zn^2+^ [F(1,56) = 8.49, *p* = 0.0051].

(2)Istradefylline and Magnesium or Zinc

As shown in Figure 3D neither Mg^2+^ nor Zn^2+^ caused statistically significant changes in *Ogg1* gene expression (*p* > 0.05) but istradefylline injected alone caused a significant decrease in the mRNA level of *Ogg1* in comparison to the control group (*p* < 0.001). Istradefylline and Mg^2+^ administered simultaneously caused a significant increase in the *Ogg1* expression in comparison to istradefylline-treated group (*p* < 0.05) and significant decrease in comparison to Mg^2+^-treated group (*p* < 0.05). Istradefylline and Zn^2+^ injected simultaneously caused significant increases the *Ogg1* mRNA level in comparison to istradefylline-treated group (*p* < 0.001).

A two-way ANOVA showed no interaction between istradefylline and Mg^2+^ [F(1,80) = 0.45, *p* = 0.5050] and a significant interaction between istradefylline and Zn^2+^ [F(1,80) = 8.51, *p* = 0.0046].

#### 2.2.6. Effect of Co-Administration of Selective Adenosine Receptor Antagonists and Magnesium or Zinc on Msra Expression

(1)DPCPX and Magnesium or Zinc

As shown in Figure 3E neither DPCPX nor Mg^2+^ nor Zn^2+^ caused statistically significant changes in *MsrA* gene expression (*p* > 0.05) as compared to the control group. DPCPX co-administered with Zn^2+^ did not cause significant changes in the mRNA level of this gene while the co-administration of DPCPX with Mg^2+^ resulted in a significant increase in *MsrA* mRNA levels both comparison to the control group (*p* < 0.05) and to the group receiving only Mg^2+^ (*p* < 0.001).

A two-way ANOVA showed no interaction between DPCPX and Mg^2+^ [F(1,56) = 3.87, *p* = 0.0542] and no interaction between DPCPX and Zn^2+^ [F(1,56) = 0.07, *p* = 0.7929].

(2)Istradefylline and Magnesium or Zinc

As shown in Figure 3E, istradefylline injected alone significantly decreased the mRNA level of *MsrA* in comparison to the control group (*p* < 0.05) while neither Mg^2+^ nor Zn^2+^ caused statistically significant changes in *MsrA* gene expression (*p* > 0.05). In the groups of animals that received istradefylline in combination with Mg^2+^ a statistically significant decrease in *MsrA* mRNA level was observed as compared to the control group (*p* < 0.001) and to the group receiving Mg^2+^ alone (*p* < 0.05). In the cerebral cortex of mice receiving istradefylline simultaneously with Zn^2+^ the statistically significant increase in the expression of the *MsrA* gene was observed as compared to the group receiving only istradefylline (*p* < 0.0001) or Zn^2+^ (*p* < 0.001).

A two-way ANOVA showed no interaction between istradefylline and Mg^2+^ [F(1,80) = 0.18, *p* = 0.6711] and a significant interaction between istradefylline and Zn^2+^ [F(1,80) = 14.96, *p* = 0.0002].

#### 2.2.7. Effect of Co-Administration of Selective Adenosine Receptor Antagonists and Magnesium or Zinc on *Nrf2* Expression

(1)DPCPX and Magnesium or Zinc

As shown in Figure 3F Zn^2+^ injected alone significantly decreased the mRNA level of *Nrf2* as compared to the control group (*p* < 0.0001). Mg^2+^ administered to mice did not cause significant changes in *Nrf2* gene expression while DPCPX significantly increased the expression of this gene (*p* < 0.0001). On the contrary, DPCPX administered simultaneously with Mg^2+^ or Zn^2+^ significantly decreased the *Nrf2* mRNA level as compared to the group of mice receiving DPCPX alone (*p* < 0.0001).

A two-way ANOVA showed a significant interaction between DPCPX and Mg^2+^ [F(1,56) = 40.43, *p* < 0.0001] and a significant interaction between DPCPX and Zn^2+^ [F(1,56) = 118.80, *p* < 0.0001].

(2)Istradefylline and Magnesium or Zinc

As shown in Figure 3F in the cerebral cortex of mice receiving istradefylline or Zn^2+^ alone the significant decrease in expression of *Nrf2* as compared to the control group was observed (*p* < 0.0001) while in the brain of mice receiving Mg^2+^ no statistically significant changes were noted. However, istradefylline and Mg^2+^ administered simultaneously caused a significant decrease in the *Nrf2* expression in comparison to the control and to the Mg^2+^-treated group (*p* < 0.0001) and a significant decrease in comparison to the Mg^2+^-treated group (*p* < 0.05). In turn, the co-administration of istradefylline and Zn^2+^ resulted in a significant increase in *Nrf2* mRNA levels as compared to the group receiving only Zn^2+^ (*p* < 0.001).

A two-way ANOVA showed no interaction between istradefylline and Mg^2+^ [F(1,80) = 3.08, *p* = 0.0832] and a significant interaction between istradefylline and Zn^2+^ [F(1,80) = 50.70, *p* < 0.0001].

## 3. Discussion

### 3.1. Behavioural Studies

In the present studies, firstly we have evaluated the synergistic antidepressant-like activity of DPCPX—selective antagonist of adenosine A1 receptor and istradefylline—selective antagonist of adenosine A2A receptor, with the magnesium and zinc ions in two commonly used despair behavioural tests (FST and TST). The FST and TST are well-characterized rodent’s tests carried out to predict the antidepressant-like efficacy of agents or agent–agent combinations [34,35]. In our research only male mice were used, in accordance with the common practice in preclinical experiments assessing the antidepressant-like effects of the active substances. The exclusion of female mice from these tests is primarily due to the presence of the oestrous cycle, thought to interfere with experimental manipulations by introducing variability, which is a less critical issue in male animals. Moreover, in order to rule out that the changes in the overall mobility of mice are not disruptive factors in these tests, a spontaneous locomotor activity test was conducted. Since no augmentation in the mice locomotor activity was observed (in the case of using Zn alone and Zn in combined with DPCPX decrease in animals’ activity was noted), the received outcomes indicate synergistic interaction between Mg^2+^/Zn^2+^ and DPCPX/istradefylline (all in ineffective doses: 10/2.5 mg/kg and 1/0.5 mg/kg, respectively) manifested in a meaningful enhancement of mice mobility in the FST and TST, which indicates antidepressant-like properties of such treatment schedule.

The same trend obtained in both conducted behavioural tests exclude the possibility of impact of various environmental factors on the presented results. In the case of the tested combination of DPCPX with Mg^2+^ or Zn^2+^, a small but statistically significant decrease in mice immobility in the FST was noted, whereas in the TST the observed difference was significantly higher. In turn, when studying the combination of istradefylline and Mg^2+^ or Zn^2+^, in both tests statistically meaningful changes were recorded at the same level. The decrease in mice immobility in the FST compared to that found in the TST is usually not identical, because these procedures are differently sensitive to various drugs. For drugs, that exhibit in the TST, while not consistently reducing immobility in the FST, include the selective serotonin reuptake inhibitors (SSRIs) [34,36]. In turn, some non-typical drugs (i.e., rolipram and levoprotillin) show antidepressant-like activity in the FST, but not in the TST [36,37], whereas shortening the immobility time in both the FST and TST is observed when using drugs, such as tricyclic antidepressants (TCAs), monoamine oxidase inhibitors (MAOIs), atypical antidepressant (i.e., mianserin) [36,38,39], and also antagonists of the NMDA [36,40,41] and agonists of the AMPA receptor [42]. As demonstrated in our previous research, A1 and A2A receptor antagonists also have antidepressant-like activity in both of these behavioural tests [31,32]. Though both these despair behavioural tests are generally based on the same principle, the neurological mechanisms underlying the observed antidepressant effect are different, which may be the reason for the observed disagreement between the antidepressant-like activity of various compounds in the FST and TST [36]. The dissimilarities in the neurotransmitter mechanisms may also be the basis of the disparities noted in our research. The adenosine A1 receptors are highly localized in the CNS, and their inhibition by DPCPX causes stimulation of voltage dependent Ca^2+^ channels, lock of K^+^ channels resulting in extenuation of hyperpolarisation and activation of adenylyl cyclase. All of them lead to an enhancement in the release of mainly monoaminergic transmitters (i.e., serotonin, noradrenalin, dopamine) [43,44,45]. Conversely, A2A receptors inhibition has no impact or decreases the release of these neurotransmitters [45]. Regarding glutamatergic transmission, the effect of A1 and A2A receptor antagonists is reversed, i.e., blockage of A1 receptors causes stimulation, and A2A receptors blockage of release of glutamate [46,47,48,49,50,51]. Additionally, the different effects of DPCPX and istradefylline on the antidepressant activity of NMDA receptor ion ligands observed in our studies may result from receptor–receptor interaction in the A1–A2A receptor complex on the striatal glutamatergic nerve terminals. Ciruela et al. [22] demonstrated that the preferential A1 receptor activation in the A1–A2A receptor complex inhibits glutamatergic neurotransmission. In turn, under conditions of high adenosine release, A2A receptor activation in the A1–A2A heteromer complex blocks mediated via A1 receptors function, and enhances release of glutamate [20,22]. Thus, the selective inhibition of A1 and A2A receptors in this complex might facilitate or inhibit glutamatergic transmission, respectively. Furthermore, Borroto–Escuela et al. [52], based on previous studies, suggest occurrence A2A–D2–NMDA heteroreceptors complexes in the striato-pallidal GABA neurons in a synaptic and/or extrasynaptic position, as well as in lower densities in cortical regions. It may, therefore, be considered that istradefylline, by inhibiting the activity of the A2A receptor, could stimulate the D2 receptor function, which inhibits NMDA receptor signalling in this heteroreceptor complex.

Several recent studies have demonstrated the synergistic interaction between adenosinergic and glutamatergic systems [30,53,54], but none of them investigated the interaction between the selective adenosine A1 and A2A receptor antagonists and magnesium ions in despair tests in mice. Likewise, in the case of the combinations of the selective A1 and A2A receptor antagonists with Zn^2+^ ions tested in presented studies, we obtained results that were in contradiction with those obtained by Lobato et al. [55]. The differences between the outcomes of behavioural tests carried out by our team and those conducted by Lobato et al. [55] may result from the use of other zinc salts and doses (zinc hydroaspartate 2.5 mg/kg and zinc chloride 30 mg/kg, respectively), other doses of DPCPX (1 mg/kg and 2 mg/kg, respectively), and alternative selective antagonist of A2A adenosine receptors at various doses (KW–6002 0.5 mg/kg and ZM241385 1 mg/kg, respectively). Additionally, the research presented in this manuscript was executed on male Albino Swiss mice, while the Lobato team performed despair test using mice of either sex. As is well-known and well-documented, gender differences can have a significant impact on depressive behaviour in rodents [55,56,57,58]. One more difference is the FST methodology, specifically the time at which the immobility of animals was evaluated. Lobato et al. [55] recorded the total duration of immobility during the whole 6 min period of behavioural testing (methodology described by Kaster et al. [59]), while in our studies the total duration of mice immobility was measured between 2 and 6 min of the experiment (methodology described by Porsolt et al. [60]).

Already in 1982 it was noted that, in the animal models, adenosine receptor agonists counteracted behavioural changes induced by of NMDA receptor antagonists [61]. Additionally, Popoli et al. [62] demonstrated that, in rats, adenosine A1 and A2A receptor agonists also prevent neurophysiological changes (i.e., the electroencephalographic effects) caused by the NMDA receptor antagonist (MK-801). In turn, Serefko et al. [30] and Bespalov et al. [53] demonstrated a synergistic interaction between a non-selective adenosine receptor antagonist, caffeine, and NMDA receptor antagonists. Obtained results indicated that, caffeine, only in the low dose ranges, given jointly with the NMDA receptor channel blocker (MK-801, 0.01–0.3 mg/kg), or a competitive antagonist (ᴅ–CPPene, 0.3–5.6 mg/kg) to rats significantly lowered brain stimulation reward thresholds [53]. Serefko and co-workers [30] demonstrated that caffeine injected concurrently at inactive dose (5 mg/kg) with the NMDA receptor antagonists, MK–801, L–701,324 or CGP 37849, also at non-effective doses (0.05, 1, 0.3 mg/kg, respectively) significantly decreased the immobility time of mice in the FST. The same antidepressant-like activity was observed in mice receiving caffeine in combination with a partial agonist of a glycine recognition site (ᴅ–cycloserine, 2.5 mg/kg) [30]. This scientific team examined also the effect of caffeine, on the antidepressant-like activity of magnesium ions in the FST in mice [30]. Unlike in the case of this study, they observed lack of a synergistic interaction between magnesium or zinc hydroaspartate and caffeine.

The explanation for interactions between adenosine and glutamatergic system indicated in our, and the other mentioned studies may be the mechanism described by Craig and White [63] and confirmed by Manzoni et al. [64] and Melani et al. [65]. Those reports showed that adenosine release in rat brain structures (i.e., cortex and striatum) is enhanced by stimulation of the NMDA receptor [63,64,65]. Contrariwise, the effects mediated via these receptors are mitigated by adenosine receptor activation [66].

### 3.2. BDNF Level Analysis

Clinical and animal studies confirm that depression is associated with neuronal atrophy and neuronal cell loss, especially in the cerebral cortex and hippocampus [67,68]. Therefore, for a better understanding of the pathogenesis of depression and the mechanisms of action of antidepressants, neurotrophic factors are considered [69]. The neurotrophic hypothesis of depression suggests that the deficiency of neurotrophic factors contributes to the pathology of the brain structures during the development of the disease, and that antidepressants may reverse this deficiency and, thus, contribute to the alleviation of symptoms of depression [70,71]. Of the many neurotrophins, most research focuses on BDNF, one of the most common neurotrophic factors in adult human and animal brains. BDNF is a protein that, in humans, is encoded by the BDNF gene; it is a member of the neurotrophin family of growth factors, which are related to the canonical nerve growth factor. BDNF is believed to play an important role in memory formation and learning [72]. Studies have shown that drugs can act as an antidepressant through BDNF–dependent neurotrophic/neuroplastic mechanisms [73].

The neurotrophic hypothesis of depression indicates that alterations in BDNF levels occur in key limbic structures to contribute to the pathogenic processes [74,75]. Up-regulation of this factor occurs in the amygdala and nucleus accumbens of persons with depression whereas down-regulation of BDNF occurs in the prefrontal cortex and hippocampus [76]. Animal studies have shown that BDNF is able to cross the blood-brain barrier in both directions and that central and peripheral BDNF levels are related [77,78]. Changes in levels of BDNF in the serum of patients with depressive disorders emphasize the potential of BDNF as a biomarker [77]. Since the levels of BDNF in the brain in patients suffering from depression are impossible to measure, the determination of the level of BDNF in the blood is possible [79]. Studies in rodents have shown that blood BDNF concentrations correlate positively with BDNF levels in the frontal cortex and hippocampus and support the notion that blood BDNF levels measurements reflect BDNF levels in brain tissue [77,80]. In our study, we did not observe statistically significant changes in BDNF levels in the serum of mice receiving magnesium or zinc hydroaspartate alone or the selective A1 or A2A receptor antagonists. However, in the groups of mice receiving DPCPX with Mg^2+^ and istradefylline with Mg^2+^ or Zn^2+^ a significant reduction in BDNF concentration was noted, both compared to the control group and to the groups with a single administration of the substances. The obtained data, especially in relation to the results of behavioural tests, may seem surprising, because, as mentioned above, one of the mechanisms of antidepressant activity includes increasing the concentration of BDNF in the brain. Perhaps the explanation for the lack of changes in the expression of this neurotrophic factor is the duration of the experiment. This prediction may be confirmed by research which has shown that up-regulation of BDNF occurs following chronic administration of antidepressants consistent with the time course for the therapeutic action of antidepressants [71,81]. Nibuya et al. [82] confirmed that chronic (21 days), but not acute (1 day), administration of several antidepressants significantly increased BDNF mRNA in the brain [82]. However, it should be emphasized that in groups of animals receiving DPCPX with Mg^2+^ and istradefylline with Mg^2+^ or Zn^2+^ a significant decrease in BDNF concentration was observed. Studies have shown that a lower BDNF level in prefrontal cortex function might modulate the synergistic effects of serotonergic and autonomic nervous system in order to maintain brain physiological and psychological homeostasis [83]. Perhaps the changes observed in our study may also be related to the above hypothesis, i.e., the desire to maintain homeostasis in the CNS during acute stress. In order to clarify this issue, further studies on the effect of the investigational drugs on the level of BDNF also after long-term administration are necessary.

### 3.3. Gene Expression Analysis

Finally, in the prefrontal cortex of mice the mRNA levels of the following genes: *Adora1* (adenosine A1 receptor), encoding A1 receptor [84], *Slc6a15* (solute carrier family 6 member 15), encoding a transporter with the neutral amino acid transporters subfamily of the *Slc6* family [85], and *Comt* (catechol–O–methyltransferase), an encoding enzyme responsible for the degradation of catecholamine neurotransmitters [86] have been measured. Moreover, the mRNA expression of DNA–repairing genes (the 8–oxoguanine glycosylase–1, *Ogg1,* and methionine sulfoxide reductase *MsrA*) and gene–encoding antioxidative transcriptional factor (nuclear factor 2–related factor 2, *Nrf2*) were determined.

In our study we have evaluated the expression of the selected genes, which may play a role in the pathophysiology and treatment of depression or the mechanism of DPCPX and istradefylline action (i.e., *Adora1, Slc6a15, Comt*). Some research shows the potential link between the improvement of the A1 receptor signalling and the antidepressant effect on people suffering from depressive disorders. *Adora1* is a gene encoding the A1 receptor. Substances that have selective impact on them may have neuroprotective function [87]. Adenosine, as an endogenous agonist of these receptors, provokes inhibition of secretion neurotransmitters from the cells, which may cause reduction of nerve cells and neuronal transmission excitability [88]. Substances that inhibit the action of adenosine contribute to an increase in the concentration of neurotransmitters and intensity of neurotransmission [89]. In our study, we did not observe significant changes in the *Adora1* gene expression in the cerebral cortex of mice receiving DPCPX or istradefylline alone. In turn, mice pretreated with Mg^2+^, Zn^2+^, DPCPX with Zn^2+^ and istradefylline with Mg^2+^ or Zn^2+^ showed an increased expression of this gene, relative to control. The observed increase in expression of this gene may be due to an increase in the concentration of neurotransmitters, since increased neurotransmission may lead to stimulation of the A1 receptor. Studies have shown that the increase of A1 expression evokes resilience against depressive-like behaviour in the behavioural tests and the antidepressant effects in animal chronic stress model [84]. Interestingly, DPCPX administered to mice in combination with Mg^2+^ causes a reduction in expression of this gene, however, when administered to mice in combination with Zn^2+^ it significantly increases its expression. Perhaps the reason for these differences are the complex mechanisms of interaction between the adenosine and glutamatergic systems associated with different types of receptors.

It is possible that the *Slc6a15* gene is associated with the major depressive disorder. The research proves the relation between chronic stress in mice and down-regulation of the *Slc6a15* in the hippocampus. Potentially, these changes lead directly to alterations in the hippocampal volume and the integrity of neurons [90]. It has been showed, that the decreased *Slc6a15* expression, due to genetic or environmental factors, might alter neuronal circuits related to the susceptibility for major depression [90]. The lower *Slc6a15* expression, especially in the hippocampus, could increase stress susceptibility, also by altering excitatory neurotransmission in the brain. The studies show that the *Slc6a15* plays a role in modulating emotional behaviour, possibly mediated by its impact on glutamatergic neurotransmission [91]. A decreased expression of this gene may lead to disturbances around the neuronal conduction, thus, the *Slc6a15* protein may be the grip point for drugs which affect its function [90]. A decreased expression of *Slc6a15* was noted in the cerebral cortex of mice receiving only istradefylline, Mg^2+^, Zn^2+^ and simultaneously istradefylline with Mg^2+^. Therefore, the reduction in the mRNA level of *Slc6a15* seems to be related to the inhibitory effect on glutamatergic transmission through antagonism of NMDA receptor complex. However, co-administration of DPCPX with Zn^2+^ and istradefylline with Zn^2+^ resulted in an increase of the mRNA level of *Slc6a15* as compared to the single drug administration, which may indicate beneficial effects. Perhaps these results may be explained by modulation of the activity of another ionotropic and metabotropic glutamate receptors by Zn^2+^.

*Comt* is one of the most important enzymes connected with the disintegration of catecholamines [92]. The enzyme accelerates the process of transferring methyl groups from S–adenosylmethionine to catecholamines. There are studies, which proved an interaction between the *Comt* gene expression and depressive disorders, whose development may be caused by some alterations of the *Comt* in the prefrontal cortex [93]. Antidepressants, by enhancing catecholamine transmission, may affect the activity of enzymes involved in their metabolism [92]. In our work, we observed an increase in *Comt* expression, as compared to controls, in groups of mice receiving only DPCPX, istradefylline, Mg^2+^ or Zn^2+^. The explanation for these changes may be a theory which assumes that these drugs cause an increase in the level of catecholamine in the brain, hence the activity of *Comt* as the enzyme responsible for the breakdown of catecholamines is higher. This could probably be a compensatory mechanism in response to higher catecholamine production. However, it should be noted that a decrease in the *Comt* gene expression was observed in the group of mice receiving DPCPX with Mg^2+^ (similar to *Adora1* expression). In order to understand this effect further research is required. In the remaining groups, no statistically significant changes in the expression of the determined gene were noted while compared to the control, however, the obtained results do not indicate a more favourable effect of combined drug administration in this respect.

In the last stage of our research the expression of selected antioxidant defence genes (i.e., *Ogg1, MsrA* and *Nrf2*) were determined. *Ogg1* gene is a key component of the base excision repair pathway; *Ogg1* encodes the enzyme responsible for excision of a mutagenic base by˗product (8–oxoguanine) that occurs as a result of exposure to reactive oxygen species (ROS) [94]. In patients with depression, increased concentrations of oxidative damage markers were detected, including 8–oxoguanine, a product of guanine oxidation, as well as malondialdehyde, a compound formed in the process of lipid peroxidation [95]. Induced by oxidative stress, the *Ogg1* gene is expressed in the brain and can be a cellular marker for oxidative DNA damage [96,97]. The current study revealed a significant decrease in the mRNA level of *Ogg1* in the prefrontal cortex of mice receiving istradefylline or DPCPX with Zn^2+^ in comparison to the control group, which may indicate potential benefits in terms of antioxidant capacity. Similarly, a lower mRNA level of *MsrA* was observed in the brain of mice receiving istradefylline or istradefylline and Mg^2+^ compared to the control group. *MsrA* plays an important role in providing cells with defence against oxidative stress. It is an enzyme that catalyses the thioredoxin-dependent reduction of free and protein–bound methionine sulfoxide (MetO) to methionine. The methionine residues bound to the protein are particularly susceptible to oxidation by ROS [98]. In addition, *MsrA* can be used to repair oxidatively damaged proteins [99]. Yeast cells and bacteria, deficient in the *MsrA* gene, are more sensitive to oxidative stress, while over-expression of *MsrA* in human T lymphocytes results in an increase in life expectancy under oxidative stress conditions. The *MsrA* protein has the highest expression in cells sensitive to oxidative damage, among others in the cerebellum and brain neurons [98]. The reported reduction in expression of DNA repair enzymes may indicate a reduction of oxidative damage and, thus, may suggest protection against oxidative stress.

A statistically significant increase in *Nrf2* gene expression was observed in the cerebral cortex of mice receiving DPCPX alone, as well as DPCPX with Mg^2+^ as compared to the control group. An increase in expression of this gene was also noted in the groups of mice treated with DPCPX with Zn^2+^ and istradefylline with Zn^2+^ as compared to Zn^2+^ alone. In turn, a reduction in *Nrf2* expression as compared to the controls was observed in the brain of mice receiving istradefylline, istradefylline with Mg^2+^ as well as Zn^2+^ alone. A reduction in the mRNA level of *Nrf2* was also observed in the group of mice receiving DPCPX in combination with Zn^2+^ as compared to DPCPX alone. The role of *Nrf2* in the mechanisms involved in neuroprotection is crucial for determining new strategies of treatment of neurodegenerative diseases. *Nrf2* is mainly activated by ROS [100]. Recent reports indicate the protective function of *Nrf2* in various pathological states, including ischemia and neurodegenerative diseases [101]. It has been proved that the nerve cells treated with chemical activators of the *Nrf2*-ARE pathway (tBHQ, sulforaphane) are more resistant to oxidative stress-induced neurotoxicity [102]. In vitro studies have also shown that astrocytes with increased expression of the *Nrf2* coding gene, resulting from adenoviral vectors, may protect the nerve cells from the effects of oxidative stress caused by hydrogen peroxide [103]. On the other hand, the lowering *Nrf2* expression results in an increased sensitivity of neurons and astrocytes to oxidative stress. This is due to the reduction of both constitutive and inducible expression of cytoprotective genes [104]. *Nrf2* is also a transcription factor that controls the expression of detoxification and antioxidant enzymes, therefore, while interpreting the above results changes in the oxidative stress response genes should be taken into account. DPCPX administered to mice does not cause any changes in the expression of *Ogg1* and *MsrA* but significantly increases the expression of *Nrf2*, what may indicate its beneficial effect. However, the observed change may also suggest an excessive increase in ROS generation, because *Nrf2* is activated by oxidative stress. Mg^2+^ administered to mice did not affect the changes of antioxidant defence gene expression. However, Zn^2+^ significantly decreased the expression of *Nrf2*, proving a potentially disadvantageous effect in this regard. The increased expression of *Nrf2* and *MsrA* resulting from administration of DPCPX together with Mg^2+^ may support the hypothesis of excess ROS and activation of the *Nrf2* pathway. In contrast, DPCPX in combination with Zn^2+^ causes a reduction in *Ogg1* expression and does not significantly affect the change of the transcription factor expression relative to control, which may indicate a beneficial influence on defence against excess ROS formation. Istradefylline administered alone decreases the expression of repair genes, suggesting a lack of oxidative DNA damage. However, given the significant reduction in the mRNA level of *Nrf2*, it can be assumed that these changes are dependent on the level of transcription factor, which induces reduced protection against oxidative stress. Similar changes were noted in the group of mice receiving istradefylline with Mg^2+^. In turn, the co-administration of istradefylline with Zn^2+^ does neither decrease the *Nrf2* expression nor change the expression of oxidative stress response genes in regard to the control, which may suggest a more favourable effect than administration of istradefylline alone.

## 4. Materials and Methods

### 4.1. Animals

Naïve 10–week–old male Albino Swiss mice (*n* = 180) weighing 25–30 g purchased from the licensed breeder (Experimental Medicine Centre, Lublin, Poland) were used for all experiments. The animals were kept in standard cages (10 mice each) in the laboratory rooms with strictly controlled housing conditions—12 h light/dark cycle (light on at 6:00 a.m. light of at 6:00 p.m.), temperature 21–23 °C, relative humidity 45–55%. Throughout the study, the animals had unrestricted access to chow pellets and tap water. To minimize circadian influences all procedures were conducted between 8 a.m. and 3 p.m. The experimental groups consisted of 8–10 mice, randomly assigned prior the drug administration. Behavioural tests were analysed by two blind observers.

### 4.2. Drugs

The following substances were used: DPCPX (8–cyclopentyl–1,3–dipropylxanthine, Sigma–Aldrich, Poznań, Poland), istradefylline (KW–6002, (E)–8–(3,4–dimethoxystyryl)– 1,3–diethyl–7–methylxanthine, Sigma–Aldrich, Poznań, Poland), magnesium hydroaspartate (Farmapol, Poznań, Poland), and zinc hydroaspartate (Farmapol, Poznań, Poland). DPCPX (1 mg/kg) and istradefylline (0.5 mg/kg) were suspended in a 0.9% saline with Tween 80 (1%) (POCH, Gliwice, Poland), whereas magnesium and zinc hydroaspartate (10 mg/kg and 2.5 mg/kg, calculated as pure magnesium or zinc ions, respectively) were dissolved in 0.9% saline. DPCPX and magnesium hydroaspartate were injected intraperitoneally (i.p.) 30 min, while istradefylline was administered orally (p.o.) and zinc hydroaspartate i.p. 60 min before behavioural testing. Animals from control group were given 0.9% saline. All liquid dosage forms were prepared immediately prior to the experiments and they were administered in a volume of 0.01 mL/g.

The treatment schedules and subtherapeutic doses of DPCPX, Mg^2+^ and Zn^2+^ were chosen on the basis of our previous projects [30,31], whereas of istradefylline were selected on the basis of literature data [105] and then confirmed in preliminary studies carried out in our laboratory.

### 4.3. Treatment Schedule

1st group: saline + saline; 2nd group: DPCPX 1 mg/kg + saline; 3rd group: istradefylline 0.5 mg/kg + saline; 4th group: Mg^2+^ 10 mg/kg + saline; 5th group: DPCPX 1 mg/kg + Mg^2+^ 10 mg/kg; 6th group: istradefylline 0.5 mg/kg + Mg^2+^ 10 mg/kg; 7th group: Zn^2+^ 2.5 mg/kg + saline; 8th group: DPCPX 1 mg/kg + Zn^2+^ 2.5 mg/kg; 9th group: istradefylline 0.5 mg/kg + Zn^2+^ 2.5 mg/kg.

### 4.4. Behavioural Studies

#### 4.4.1. Forced Swim Test (FST)

The FST procedure was conducted according to the method described by Porsolt et al. [60]. Each animal was placed singly into the glass cylinders (height 25 cm, diameter 10 cm) containing 10 cm of water at 23–25 °C for 6 min. Since slight immobility was observed during the first 2 min, the total duration of immobility was recorded between the 2nd and the 6th min of the experiment. The animal was considered immobile when it stopped struggling and remained motionless, making only the movements indispensable to keep head above the water.

The results obtained in the FST were presented as the arithmetic mean of immobility time of mice in s ± standard error of the mean (SEM) for each group.

#### 4.4.2. Tail Suspension Test (TST)

The TST procedure was conducted according to the method described by Steru et al. [39]. Each animal was suspended singly by the tail using adhesive tape for 6 min. Since slight immobility was observed during the first 2 min, the total duration of immobility was recorded between the 2nd and the 6th min of the experiment. The animal was considered immobile when it stopped struggling and ceased moving limbs and body, making only the movements indispensable to breathe.

The results obtained in the TST were presented as the arithmetic mean of immobility time of mice in s ± standard error of the mean (SEM) for each group.

#### 4.4.3. Spontaneous Locomotor Motility

The spontaneous locomotor motility was assessed using an Opto-Varimex-4 Auto–Track (Columbus Instruments, USA). The actimeter consists of four transparent cages (43 × 43 × 32 cm) covered with lids, a set of four infrared emitters, with 16 laser beams each, and four detectors tracking and monitoring mice movements. After administration of respective drug/drug combinations each animal was placed individually into the cages for 6 min. Spontaneous locomotor motility was analysed between the 2nd and the 6th min of the experiment, which corresponds with the time interval evaluated in the FST and TST.

The results obtained in the spontaneous locomotor motility test were presented as the arithmetic average distance travelled (in cm) by animals ± SEM for each group.

### 4.5. Biochemical and Molecular Studies

#### 4.5.1. BDNF Levels Analysis

(1)Collection of Blood

After the behavioural tests, the mice were decapitated by experienced animal technicians with the appropriate certificates. The murine blood was collected into Eppendorf tubes and allowed to clot at room temperature. Then, the blood was centrifuged for 10 min at 5000× *g* and serum was collected into polyethylene tubes and frozen at −25 °C.

(2)Determination of BDNF Concentration

The concentration of BDNF in murine serum was measured by a ready–to–use sandwich enzyme immunoassay (ELISA) diagnostic kit dedicated to mouse fluids and tissues (Enzyme–linked Immunosorbent Assay Kit For BDNF, Cloud–Clone Corp., Katy, TX, USA). All procedures were conducted according to the manufacturer’s instructions.

#### 4.5.2. Gene Expression Analysis

(1)Collection of Prefrontal Cortex

After the behavioural tests, the mice were decapitated by an experienced animal technicians with the appropriate certificates. The brain of mice was carefully removed immediately after the decapitation and immersed in cooled (2–8 °C) saline to remove blood. The prefrontal cortex was isolated and washed with 20 µL injection solution and stored in a freezer at −80 °C for biochemical and molecular studies.

(2)RNA Isolation

Total RNA was isolated from 30 mg of murine prefrontal cortex using TRIzol Reagent (Invitrogen, Carlsbad, CA, USA) according to the manufacturer’s instructions. Briefly, 500 µL of TRIzol reagent was added to the tissue and homogenized using a homogenizer. Then, 100 µL of chloroform (POCH, Gliwice, Poland) was added, and the mixture was incubated for 3 min and centrifuged (15 min, 12,000× *g*, 4 °C). The colourless aqueous phase containing RNA was transferred to a new tube and 250 µL of isopropanol (POCH, Gliwice, Poland) was added. The obtained mixture was incubated for 10 min and then centrifuged (10 min, 12,000× *g*, 4 °C). The formed white pellet at a bottom of the tube was washed with 500 µL of 75% ethanol (POCH, Gliwice, Poland), dried and dissolved in 50 µL of RNAse–free water (EURx, Gdańsk, Poland).

The concentration and purity of RNA were measured spectrophotometrically using a NanoDrop Maestro Nano spectrophotometer (Maestrogen, Hsinchu, Taiwan). High purity RNA was used for further investigations (A260/280 ratio ranged between 1.8 and 2.0).

(3)cDNA Synthesis

Synthesis of cDNA was performed using a high-capacity cDNA reverse transcription kit (Applied Biosystems, Foster City, CA, USA) according to the manufacturer’s procedure. Briefly, the following reaction mixture was prepared in triplicates: 2 µL of 10X RT Buffer, 0.8 µL of 25X dNTP Mix (100 mM), 2 µL of 10X RT random primers, 1 µL of MultiScribe reverse transcriptase (50 U/µL), 0.5 µL of RNase inhibitor (40 U/µL), 10 µL of isolated RNA (200 ng/µL) and 3.2 µL of RNase-free water.

The reaction conditions: 25 °C for 10 min, 37 °C for 120 min, and then at 85 °C for 5 min to complete the process. Received cDNA was stored at −20 °C.

(4)Real-Time PCR

The relative expression of following genes was measured: *Ogg1, Msra, Nfe2l2, Adora1, Comt* and *Slc6a15* by real–time PCR reaction, ΔΔCt method, using *Hprt* and *Tbp* as endogenous controls. The reaction was conducted in triplicate using the 7500 Fast Real–Time PCR System (Applied Biosystems, Foster City, CA, USA) and Fast Probe qPCR Master Mix (2×), plus ROX Solution (EURx, Gdańsk, Poland). Briefly, reaction mixture contained 10 µL of Fast Probe qPCR Master Mix (2×), 9 µL of RNase–free water, 0.5 µL of ROX Solution (50 nM), and 0.5 µM of gene–specific TaqMan probe (Applied Biosystems, Foster City, CA, USA) described in Table 2. The reactions was performed as followed: 95 °C for 3 min, 40 cycles: 95 °C for 10 s and 60 °C for 30 s. The data quality screen based on amplification, Tm and Ct values was performed to remove any outlier data before ΔΔCt calculations and to determine fold change in mRNA levels. The outcomes were shown as RQ value (RQ = 2^–ΔΔCt^).

### 4.6. Statistical Analysis

A two-way ANOVA followed by a Bonferroni’s post hoc test was used to analyse the results of the behavioural, biochemical and molecular studies and to determine interactions occurring between tested drugs when administered jointly. The differences between groups were considered statistically significant when *p* < 0.05.

## 5. Conclusions

Summarizing, outcomes introduced in this research clearly show positive interaction of selective A1 and A2A adenosine receptor antagonists with Mg^2+^ and Zn^2+^, manifested as an antidepressant-like effect in the FST and TST without any stimulation of the animals’ locomotor activity. This may indicate an interrelationship between adenosine and glutamatergic system, which could be responsible for the observed antidepressant activity during behavioural examination. In connection to the behavioural tests the biochemical results are surprising, because reducing the BDNF level in murine serum after co-administration of both selective antagonists of A1 and A2A receptors with the potent ion antagonists of the NMDA receptor complex was noted. Therefore, further research is needed to confirm and explain these changes. Additionally, molecular studies showed that (1) the pre-treatment with Mg^2+^ or Zn^2+^, both alone and in the combination with DPCPX as well as with istradefylline causes changes in A1 receptor expression, (2) co-administration of DPCPX with Zn^2+^ and istradefylline with Zn^2+^ results in an increase the mRNA level of *Slc6a15* as compared to the single-drug administration, and (3) co-administration of tested agents does not have a more favourable effect on the *Comt* gene expression. Moreover, the changes obtained in the expression of selected antioxidant defence genes indicate that joint treatment with DPCPX-Mg^2+^, DPCPX-Zn^2+^, istradefylline-Mg^2+^ and istradefylline-Zn^2+^ may have a more antioxidant capacity benefits than administration of DPCPX and istradefylline alone. These mechanisms might be responsible for the observed profitable effect of such concomitant treatment in mice despair tests. It seems plausible that a combination of selective A1 as well as A2A receptor antagonist and magnesium or zinc may be a new strategy to be used in the treatment of patients suffering from depressive disorders.

## Figures and Tables

**Figure 1 ijms-22-01840-f001:**
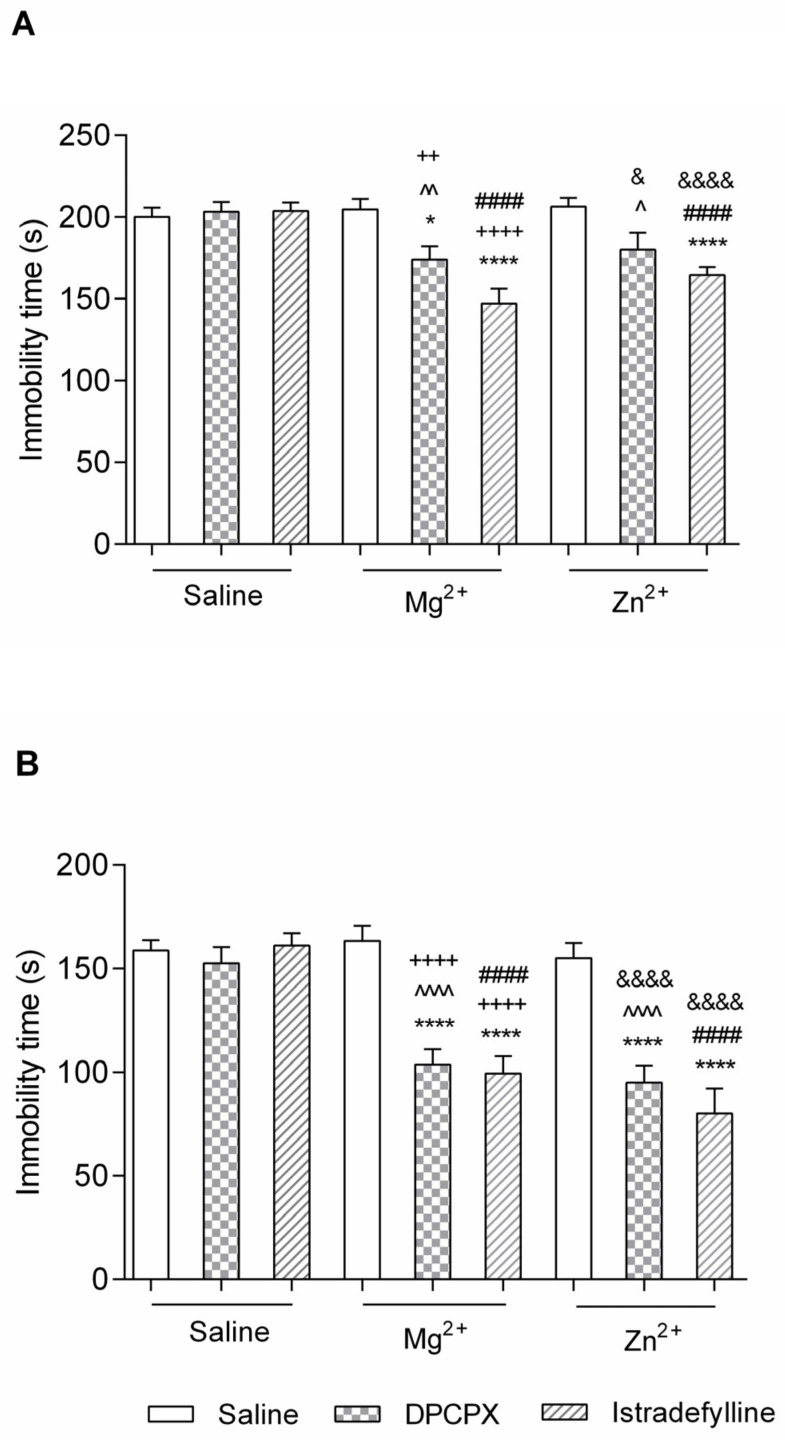
Effect of co-administration of DPCPX and istradefylline with Mg^2+^ and Zn^2+^ in the (**A**) FST and (**B**) TST in mice. DPCPX, Mg^2+^ and saline were administered i.p. 30 min, whereas istradefylline p.o. and Zn^2+^ i.p. 60 min prior behavioural testing. The data are presented as the means ± SEM. Each experimental group consisted of 10 animals. (**A**) * *p* < 0.05, **** *p* < 0.0001 vs. NaCl-treated group; ^ *p* < 0.05, ^^ *p* < 0.01 vs. DPCPX-treated group; ^####^
*p* < 0.0001 vs. istradefylline-treated group; ^++^
*p* < 0.01 vs. Mg^2+^-treated group; ^&^
*p* < 0.05, ^&&&&^
*p* < 0.0001 vs. Zn^2+^-treated group (two-way ANOVA followed by Bonferroni’s post hoc test); (**B**) **** *p* < 0.0001 vs. NaCl-treated group; ^^^^ *p* < 0.0001 vs. DPCPX-treated group; ^####^
*p* < 0.0001 vs. istradefylline-treated group; ^++++^
*p* < 0.0001 vs. Mg^2+^-treated group; ^&&&&^
*p* < 0.0001 vs. Zn^2+^-treated group (two-way ANOVA followed by Bonferroni’s post hoc test)

**Figure 2 ijms-22-01840-f002:**
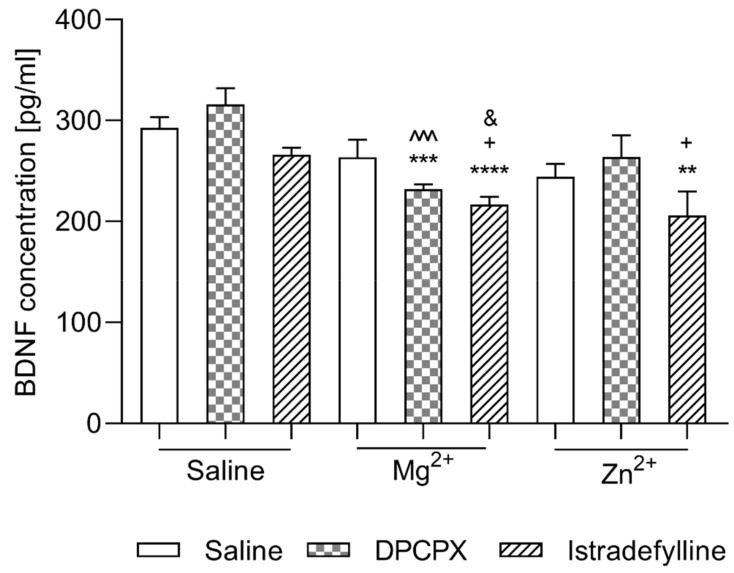
Effect of co-administration of DPCPX and istradefylline with Mg^2+^ and Zn^2+^ on BDNF concentration in mice serum. DPCPX, Mg^2+^ and saline were administered i.p. 30 min, whereas istradefylline p.o. and Zn^2+^ i.p. 60 min prior decapitation. The data are presented as the means ± SEM. Each experimental group consisted of 10 animals. ** *p* < 0.01, *** *p* < 0.001, **** *p* < 0.0001 vs. NaCl-treated group; ^^^ *p* < 0.001 vs. DPCPX- treated group; ^+^
*p* < 0.05 vs. Mg^2+^-treated group; ^&^
*p* < 0.05 vs. Zn^2+^-treated group (two-way ANOVA followed by Bonferroni’s post hoc test).

**Figure 3 ijms-22-01840-f003:**
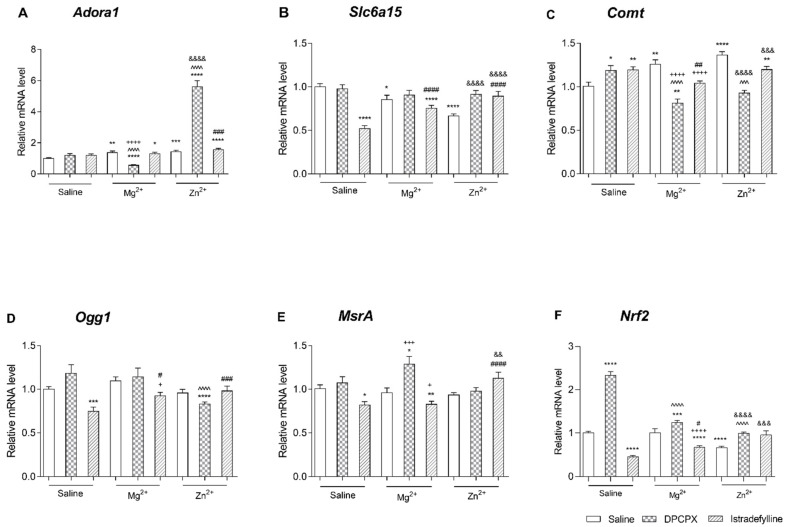
Effect of co-administration of DPCPX and istradefylline with Mg^2+^ and Zn^2+^ on gene expression (**A**) *Adora1*, (**B**) *Slc6a15*, (**C**) *Comt*, (**D**) *Ogg1*, (**E**) *MsrA*, and (**F**) *Nrf2* in mice prefrontal cortex. DPCPX, Mg^2+^ and saline were administered i.p. 30 min, whereas istradefylline p.o. and Zn^2+^ i.p. 60 min prior decapitation. The data are presented as the means ± SEM. Each experimental group consisted of 10 animals. (**A**) * *p* < 0.05, ** *p* < 0.01, *** *p* < 0.001, **** *p* < 0.0001 vs. NaCl-treated group; ^^^^ *p* < 0.0001 vs. DPCPX-treated group; ^###^
*p* < 0.001 vs. istradefylline-treated group; ^++++^
*p* < 0.0001 vs. Mg^2+^-treated group; ^&&&&^
*p* < 0.0001 vs. Zn^2+^-treated group (two-way ANOVA followed by Bonferroni’s post hoc test). (**B**) * *p* < 0.05, **** *p* < 0.0001 vs. NaCl-treated group; ^####^
*p* < 0.0001 vs. istradefylline-treated group; ^&&&&^
*p* < 0.0001 vs. Zn^2+^-treated group (two-way ANOVA followed by Bonferroni’s post hoc test). (**C**) * *p* < 0.05, ** *p* < 0.01, **** *p* < 0.0001 vs. NaCl-treated group; ^^^ *p* < 0.001, ^^^^ *p* < 0.0001 vs. DPCPX-treated group; ^#^
*p* < 0.05, ^##^
*p* < 0.01 vs. istradefylline-treated group; ^++++^
*p* < 0.0001 vs. Mg^2+^-treated group; ^&&&^
*p* < 0.001, ^&&&&^
*p* < 0.0001 vs. Zn^2+^-treated group (two-way ANOVA followed by Bonferroni’s post hoc test). (**D**) *** *p* < 0.001, **** *p* < 0.0001 vs. NaCl-treated group; ^^^^ *p* < 0.0001 vs. DPCPX-treated group; ^#^
*p* < 0.05, ^###^
*p* < 0.001 vs. istradefylline-treated group; ^+^
*p* < 0.05 vs. Mg^2+^-treated group (two-way ANOVA followed by Bonferroni’s post hoc test). (**E**) * *p* < 0.05, ** *p* < 0.01 vs. NaCl-treated group; ^####^
*p* < 0.0001 vs. istradefylline-treated group; ^+^
*p* < 0.05, ^+++^
*p* < 0.001 vs. Mg^2+^-treated group; ^&&^
*p* < 0.01 vs. Zn^2+^-treated group (two-way ANOVA followed by Bonferroni’s post hoc test). (**F**) *** *p* < 0.001, **** *p* < 0.0001 vs. NaCl-treated group; ^^^^ *p* < 0.0001 vs. DPCPX- treated group; ^#^
*p* < 0.05 vs. istradefylline-treated group; ^++++^
*p* < 0.0001 vs. Mg^2+^-treated group; ^&&&^
*p* < 0.001, ^&&&&^
*p* < 0.0001 vs. Zn^2+^-treated group (two-way ANOVA followed by Bonferroni’s post hoc test).

**Table 1 ijms-22-01840-t001:** Effect of treatments on mice spontaneous motility.

Treatment (mg/kg)	Distance (cm) between the 2nd and the 6th Minute
saline + saline	1196.5 ± 58.3
DPCPX 1 + saline	1111.8 ± 57.1
istradefylline 0.5 + saline	1093.2 ± 54.7
Mg^2+^ 10 + saline	1028.9 ± 73.2
DPCPX 1 + Mg^2+^ 10	905.40 ± 93.5
istradefylline 0.5 + Mg^2+^ 10	1150.9 ± 37.3
Zn^2+^ 2.5 + saline	986.90 ± 38.2 *
DPCPX 1 + Zn^2+^ 2.5	766.40 ± 89.7 ***^,^^^^^,&^
istradefylline 0.5 + Zn^2+^ 2.5	1186.8 ± 90.9

DPCPX, Mg^2+^ and saline were administered i.p. 30 min, whereas istradefylline p.o. and Zn^2+^ i.p. 60 min prior spontaneous motility test. Distance travelled was recorded between the 2nd and the 6th min of the test. Each experimental group consisted of 9–10 animals. Data are presented as the means ± SEM. * *p* < 0.05, *** *p* < 0.001 vs. NaCl-treated group; ^^^ *p* < 0.001 vs. DPCPX-treated group; ^&^
*p* < 0.05 vs. Zn^2+^-treated group (two-way ANOVA followed by Bonferroni’s post hoc test).

**Table 2 ijms-22-01840-t002:** Gene symbols, gene names, GenBank reference sequence accession numbers, assay IDs and amplicon lengths (bp).

Gene Symbol	Gene Name	Ref. Seq	Assay ID	Amplicon Length
*Ogg1*	8–Oxoguanine DNA–glycosylase 1	NM_010957.4	Mm00501784_m1	90
*Msra*	Methionine sulfoxide reductase A	NM_001253712.1 NM_001253714.1 NM_001253716.1 NM_026322.4	Mm00452737_m1	69
*Nfe2l2*	Nuclear factor, erythroid derived 2, like 2	NM_010902.3	Mm00477784_m1	61
*Adora1*	Adenosine A1 receptor	NM_001008533.3 NM_001039510.2 NM_001282945.1	Mm01308023_m1	58
*Slc6a15*	Solute carrier family 6 (neurotransmitter transporter), member 15	NM_001252330.1 NM_175328.3	Mm00558415_m1	84
*Comt*	Catechol–*O*–methyl– transferase	NM_001111062.1 NM_001111063.1 NM_007744.3	Mm00514377_m1	97
*Hprt*	Hypoxanthine guanine phosphoribosyl transferase	NM_013556.2	Mm00446968_m1	65
*Tbp*	TATA box binding protein	NM_013684.3	Mm00446974_m1	105

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
