# Peer review of "The Interaction of Selective A1 and A2A Adenosine Receptor Antagonists with Magnesium and Zinc Ions in Mice: Behavioural, Biochemical and Molecular Studies"

_ijms, 2021, doi:10.3390/ijms22041840_

Round 1

Reviewer 1 Report

Manuscript described the behavioral, biochemical and molecular studies on the interaction of selective A1 and A2A adenosine receptor antagonists with magnesium and zinc ions in mice. The objective of study was to investigate whether the co–administration of Mg2+ and Zn2+ with selective A1 and A2A receptor antagonist as interesting antidepressant strategy.

The authors having an interest in the effect of simultaneous administration of magnesium and zinc hydroaspartate with the selective A1 or A2A receptor antagonist (DPCPX 92 and istradefylline. Although, there is a lot evidence for the involvement mentioned above ions in the etiopathogenesis and therapy of depressive disorders, their pharmacological mechanism of action is still not clear. Originality of presented research is based of premise that the interaction between selective adenosine A1 or A2A receptor antagonists and ionic NMDA receptor antagonists may be a new strategy for use in the treatment of patient suffering from depressive disorders.

The results are interpreted properly and all conclusions are justified and supported by the results. Moreover, hypotheses and speculations are carefully identified. Despite the interest of such a project and the quality of the study, there are minor concerns that should be addressed:

  1. Selection of the doses of drugs antagonists, magnesium hydroaspartate and zinc hydroaspartate, as well. No information about criteria of selection. Authors should justify their choice.
  2. In the introduction and discussion the issue of adenosine heteroreceptor complexes should be statement. Moreover, distribution of AR in CNS (synaptic and extrasynaptic positions at the pre and postsynaptic level, astroglia and microglia, striato-pallidal GABA neurons).
  3. In my opinion BDNF level analysis should be delated – conclusions are speculative matter.

Minor points:

  • I encourage the authors to carefully checking sentences in terms of their length. Some of them are too long. Please divide such sentences (page 2,4
  • Results – Figures - Joint administration? Another synonym? Over the abstract and text there is simultaneous administration, concomitant administration, co-administration, combination etc. Please unify terminology in the results part.

Author Response

Manuscript described the behavioral, biochemical and molecular studies on the interaction of selective A1 and A2A adenosine receptor antagonists with magnesium and zinc ions in mice. The objective of study was to investigate whether the co–administration of Mg2+ and Zn2+ with selective A1 and A2A receptor antagonist as interesting antidepressant strategy.

The authors having an interest in the effect of simultaneous administration of magnesium and zinc hydroaspartate with the selective A1 or A2A receptor antagonist (DPCPX 92 and istradefylline. Although, there is a lot evidence for the involvement mentioned above ions in the etiopathogenesis and therapy of depressive disorders, their pharmacological mechanism of action is still not clear. Originality of presented research is based of premise that the interaction between selective adenosine A1 or A2A receptor antagonists and ionic NMDA receptor antagonists may be a new strategy for use in the treatment of patient suffering from depressive disorders.

The results are interpreted properly and all conclusions are justified and supported by the results. Moreover, hypotheses and speculations are carefully identified. Despite the interest of such a project and the quality of the study, there are minor concerns that should be addressed:

Selection of the doses of drugs antagonists, magnesium hydroaspartate and zinc hydroaspartate, as well. No information about criteria of selection. Authors should justify their choice.

The treatment schedules and subtherapeutic doses of DPCPX, Mg2+ and Zn2+ were chosen on the basis of our previous projects [Serefko et al. 2016; Szopa et al. 2018], whereas those of istradefylline were selected on the basis of literature data [Yamada et al. 2013] and then confirmed in preliminary studies carried out in our laboratory (not presented data). In accordance with the Reviewer's comments this information has been added in the subsection 4.2. Drugs.

  • Serefko, A.; Szopa, A.; Wlaź, A.; WoÅ›ko, S.; Wlaź, P.; Poleszak, E. Synergistic antidepressant-like effect of the joint administration of caffeine and NMDA receptor ligands in the forced swim test in mice. Neural Transm. (Vienna.) 2016, 123 (4), 463-472.
  • Szopa, A.; Poleszak, E.; Bogatko, K.; Wyska, E.; WoÅ›ko, S.; Doboszewska, U.; ÅšwiÄ…der, K.; Wlaź, A.; Dudka, J.; Wróbel, A.; Wlaź, P.; Serefko, A. DPCPX, a selective adenosine A1 receptor antagonist, enhances the antidepressant-like effects of imipramine, escitalopram, and reboxetine in mice behavioral tests. Naunyn Schmiedebergs Arch. Pharmacol. 2018, 391 (12), 1361-1371.
  • Yamada, K.; Kobayashi, M.; Mori, A.; Jenner, P.; Kanda, T. Antidepressant-like activity of the adenosine A(2A) receptor antagonist, istradefylline (KW-6002), in the forced swim test and the tail suspension test in rodents. Biochem. Behav. 2013, 114-115, 23-30.
  1. In the introduction and discussion the issue of adenosine heteroreceptor complexes should be statement. Moreover, distribution of AR in CNS (synaptic and extrasynaptic positions at the pre and postsynaptic level, astroglia and microglia, striato-pallidal GABA neurons).

In line with the Reviewer's comments we have added information about distribution of A1 and A2A receptors in the CNS in the Introduction section, while the issue of adenosine heteroreceptor complexes was imparted in the Introduction and Discussion section.

  1. In my opinion BDNF level analysis should be delated – conclusions are speculative matter.

Having considered this remark, the authors have decided not to discard the results of the BDNF analysis. We agree with the Reviewer's opinion that the conclusions are speculative, but we underline this in the manuscript. Moreover,  we have added the sentence: „Further research is needed to confirm and explain these changes” in the Conclusions section. We believe that the obtained results, whatever they are, should be published, since they may inspire other researchers to undertake research in this area. Often, the obtained research results make us continue research on a certain issue in further studies. Therefore, we believe that the current achievements should be subject to publication.

Minor points:

  • I encourage the authors to carefully checking sentences in terms of their length. Some of them are too long. Please divide such sentences (page 2,4)

As suggested by the Reviewer the manuscript has been revised by a native speaker.

  • Results – Figures - Joint administration? Another synonym? Over the abstract and text there is simultaneous administration, concomitant administration, co-administration, combination etc. Please unify terminology in the results part.

In accordance with the Reviewer's comments, the terminology in the manuscript has been unified. The term “co-administration” was used instead of “concomitant administration”, “joint administration” and “simultaneous administration”.

Reviewer 2 Report

The manuscript: "The interaction of selective A1 and A2A adenosine receptor antagonists with magnesium and zinc ions in mice: behavioral, biochemical and molecular studies " by Aleksandra Szopa and colleagues investigate the antidepressant efficiency of  Mg2+ and Zn2+ co-administration with selective A1 and A2A receptor antagonist in a mouse model. Previously, antidepressant role of Mg2+ and Zn2+ has been studied in different clinical trials. These studies have suggested Mg2+ and Zn2+ being potentially effective for mild-to-moderate depression in adults. Hence, the results and conclusion of the submitted study are in line with previously established notion. In contrast to clinical trials, the present study not only analyzes the behavioral aspects but also perform the subsequent biochemical and molecular analyses.  The experiments are elaborately performed and the manuscript is well-written with promising results giving a good overview of the study.

After thoroughly going through the manuscript, I have some minor comments:

  1. Please mention the age of the mice used in the experiment. In the manuscript it is mentioned that the adult mice were used. Were all the mouse of same age or was there any difference in the age. The difference of even a week between the ages of mice in different groups might impart biasness.
  2. Why were only male mice used for the experiment? Gender difference is known to impart difference in behavioral patterns in humans as well as in mice. Please address this point in the manuscript.
  3. Most of the clinical studies involving different compounds administration in small animals generally show positive outcome due to monitored and controlled experimental settings. I would like to know the opinion (speculation) of the authors, whether the co-administration of Mg2+ and Zn2+ will show similar positive behavioral, biochemical and molecular effects in humans as in mice.
  4. Please consider double checking the grammatical and syntax errors in the manuscript. For example: Page 2, line: 62: It was showed …… should be It was shown…; same page line: 69: also is used in false position. These are only a couple of examples; there are a lot of such discrepancies in the manuscript.

Author Response

The manuscript: "The interaction of selective A1 and A2A adenosine receptor antagonists with magnesium and zinc ions in mice: behavioral, biochemical and molecular studies " by Aleksandra Szopa and colleagues investigate the antidepressant efficiency of  Mg2+ and Zn2+ co-administration with selective A1 and A2A receptor antagonist in a mouse model. Previously, antidepressant role of Mg2+ and Zn2+ has been studied in different clinical trials. These studies have suggested Mg2+ and Zn2+ being potentially effective for mild-to-moderate depression in adults. Hence, the results and conclusion of the submitted study are in line with previously established notion. In contrast to clinical trials, the present study not only analyzes the behavioral aspects but also perform the subsequent biochemical and molecular analyses. The experiments are elaborately performed and the manuscript is well-written with promising results giving a good overview of the study.

After thoroughly going through the manuscript, I have some minor comments:

  1. Please mention the age of the mice used in the experiment. In the manuscript it is mentioned that the adult mice were used. Were all the mouse of same age or was there any difference in the age. The difference of even a week between the ages of mice in different groups might impart biasness.

Since the difference in the age of animals can have a significant impact on the behavioral test results, all mice used in our experiments were the same age, i.e. 10-week-old and weighing 25-30 g. In accordance with the Reviewer's comments, we have added this information in subsection 4.1. Animals.

  1. Why were only male mice used for the experiment? Gender difference is known to impart difference in behavioral patterns in humans as well as in mice. Please address this point in the manuscript.

In the presented studies only male Albino Swiss mice were used, in accordance with the common practice in preclinical experiments assessing the antidepressant-like effects of the active agents. The exclusion of female mice from preclinical studies is primarily due to the presence of the oestrous cycle, thought to interfere with experimental manipulations by introducing variability, which is a less critical issue in male animals. It would be interesting to repeat the same studies in female mice population and compare the results with those obtained in the present study.

In line with the Reviewer's remark, the rationale for selecting only male mice for the presented experiment was provided in Discussion section.

  1. Most of the clinical studies involving different compounds administration in small animals generally show positive outcome due to monitored and controlled experimental settings. I would like to know the opinion (speculation) of the authors, whether the co-administration of Mg2+ and Zn2+ will show similar positive behavioral, biochemical and molecular effects in humans as in mice.

Antidepressant activity of Mg2+ and Zn2+ was extensively studied in preclinical studies and the obtained results clearly indicate antidepressant potential of both ions. Likewise, clinical research also confirms the effectiveness of both Mg2+ and Zn2+ ions in the treatment of depression (for review see Serefko et al. 2013; Doboszewska et al. 2017; Szewczyk et al. 2018). Similarly, both non-selective and selective adenosine receptor antagonists are used successfully in alleviating symptoms and/or treating various central nervous system disorders, including Parkinson's, Alzheimer's disease, depression) (for review see Stone et al. 2009; van Calker et al. 2019). In our opinion, there is a good chance that co-administration of Mg2+ and Zn2+ with adenosine receptor antagonists in human will show similar positive behavioural, biochemical as well as molecular effects as in rodents. Therefore, we are convinced that the combined use of a selective A1 or A2A adenosine receptor antagonist in combination with Mg2+ and/or Zn2+ supplementation may in the future become a new therapeutic strategy for patients suffering from the central nervous system diseases, including depression. We hope that the outcomes of our research may inspire further clinical studies on the role of interaction between adenosinergic and glutaminergic system in the pathogenesis and therapy of depression. We suppose that this topic is worth further and more detailed preclinical and clinical studies.

Reviewer 3 Report

The present study titled “The interaction of selective A1 and A2A adenosine receptor antagonists with magnesium and zinc ions in mice: behavioral, biochemical and molecular studiesis presented in detailed well-elaborated research which revealed the activity of DPCPX–Mg2+, DPCPX–Zn2+,  istradefylline–Mg2+, and istradefylline–Zn2+ co-treatment, as a new anti-depressant therapeutic strategy.

Firstly, the authors reported on the results obtained after the behavioral studies (FST, TST) performed on male mice treated with diverse drugs. Thereafter, the authors investigated the biochemical and molecular studies.

In general, the data are strong, with suitable controls, and convincingly shows that the ionic metals, Mg2+ or Zn 2+   co-administrated with DPCPX  or istradefylline lead to an antidepressant-like effect on studies mice. The manuscript is well written, concise and the appropriate analyses are performed.

Overall, this is a well-performed study that I consider that is important and represent a new strategy to conveniently obtain a new strategy for the treatment of patients suffering from depressive disorders.

The authors need to address the below comments to strengthen the quality of the manuscript:

  1. Please explain the choice of used concentration of tested substances and drugs in the present study.
  2. Improve the English [syntax, spelling, typos mistakes: line 166, line 466 (reference 56), line 566].

Author Response

The present study titled “The interaction of selective A1 and A2A adenosine receptor antagonists with magnesium and zinc ions in mice: behavioral, biochemical and molecular studies” is presented in detailed well-elaborated research which revealed the activity of DPCPX–Mg2+, DPCPX–Zn2+,  istradefylline–Mg2+, and istradefylline–Zn2+ co-treatment, as a new anti-depressant therapeutic strategy.

Firstly, the authors reported on the results obtained after the behavioral studies (FST, TST) performed on male mice treated with diverse drugs. Thereafter, the authors investigated the biochemical and molecular studies.

In general, the data are strong, with suitable controls, and convincingly shows that the ionic metals, Mg2+ or Zn 2+   co-administrated with DPCPX  or istradefylline lead to an antidepressant-like effect on studies mice. The manuscript is well written, concise and the appropriate analyses are performed.

Overall, this is a well-performed study that I consider that is important and represent a new strategy to conveniently obtain a new strategy for the treatment of patients suffering from depressive disorders.

 The authors need to address the below comments to strengthen the quality of the manuscript:

  1. Please explain the choice of used concentration of tested substances and drugs in the present study.

The treatment schedules and subtherapeutic doses of DPCPX, Mg2+ and Zn2+ were chosen on the basis of our previous projects [Serefko et al. 2016; Szopa et al. 2018], whereas those of istradefylline were selected on the basis of literature data [Yamada et al. 2013] and then confirmed in preliminary studies carried out in our laboratory (not presented data). This information has been added in the subsection 4.2. Drugs.

  • Serefko, A.; Szopa, A.; Wlaź, A.; WoÅ›ko, S.; Wlaź, P.; Poleszak, E. Synergistic antidepressant-like effect of the joint administration of caffeine and NMDA receptor ligands in the forced swim test in mice. Neural Transm. (Vienna.) 2016, 123 (4), 463-472.
  • Szopa, A.; Poleszak, E.; Bogatko, K.; Wyska, E.; WoÅ›ko, S.; Doboszewska, U.; ÅšwiÄ…der, K.; Wlaź, A.; Dudka, J.; Wróbel, A.; Wlaź, P.; Serefko, A. DPCPX, a selective adenosine A1 receptor antagonist, enhances the antidepressant-like effects of imipramine, escitalopram, and reboxetine in mice behavioral tests. Naunyn Schmiedebergs Arch. Pharmacol. 2018, 391 (12), 1361-1371.
  • Yamada, K.; Kobayashi, M.; Mori, A.; Jenner, P.; Kanda, T. Antidepressant-like activity of the adenosine A(2A) receptor antagonist, istradefylline (KW-6002), in the forced swim test and the tail suspension test in rodents. Biochem. Behav. 2013, 114-115, 23-30.
  •  
  1. Improve the English [syntax, spelling, typos mistakes: line 166, line 466 (reference 56), line 566].

As suggested by the Reviewer the manuscript has been revised by a native speaker.